# One-Pot Fabrication of Nanocomposites Composed of Carbon Nanotubes and Alumina Powder Using a Rotatable Chemical Vapor Deposition System

**DOI:** 10.3390/ma16072735

**Published:** 2023-03-29

**Authors:** Jong-Hwan Lee, Hyun-Ho Han, Jong-Min Seo, Goo-Hwan Jeong

**Affiliations:** Department of Advanced Materials Science and Engineering, Kangwon National University, Chuncheon 24341, Gangwon-do, Republic of Korea

**Keywords:** rotatable chemical vapor deposition, one-pot synthesis, nanocomposites, carbon nano tubes, alumina, powders

## Abstract

The fabrication of multi-dimensional nanocomposites has been extensively attempted to achieve synergistic performance through the uniform mixing of functional constituents. Herein, we report a one-pot fabrication of nanocomposites composed of carbon nanotubes (CNTs) and Al_2_O_3_ powder. Our strategy involves a synthesis of CNTs on the entire Al_2_O_3_ surface using a rotatable chemical vapor deposition system (RCVD). Ehylene and ferritin-induced nanoparticles were used as the carbon source and wet catalyst, respectively. The RCVD was composed of a quartz reaction tube, 5.08 cm in diameter and 150 cm in length, with a rotation speed controller. Ferritin dissolved in deionized water was uniformly dispersed on the Al_2_O_3_ surface and calcinated to obtain iron nanoparticles. The synthesis temperature, time, and rotation speed of the chamber were the main parameters used to investigate the growth behavior of CNTs. We found that the CNTs can be grown at least around 600 °C, and the number of tubes increases with increasing growth time. A faster rotation of the chamber allows for the uniform growth of CNT by the tip-growth mechanism. Our results are preliminary at present but show that the RCVD process is sufficient for the fabrication of powder-based nanocomposites.

## 1. Introduction

Since the discovery of carbon nanotubes (CNTs) [1], extensive studies have been carried out to understand their potential in industrial applications [2,3,4,5]. Based on these studies, it is understood that the growth of CNTs with defined structure, morphology, and quantity is critical for such applications. For example, chirality-controlled growth of high-quality single-walled CNTs (SWNTs) is essential for the fabrication of nanoscale electronics [6,7,8,9,10,11,12]. In addition, mass production of moderate-quality CNTs is vital for the development of CNT-based composites, which find applications in energy storage, thermal management, electromagnetic wave shielding, etc [13,14,15,16,17,18,19,20,21]. 

Chemical vapor deposition (CVD) has been widely used to synthesize CNTs owing to its relatively simple apparatus and ease of process control [6,7,8,9,10,11,12,13]. A horizontal CVD is used for high quality CNT growth. In contrast, a vertical-type CVD unit has been employed to achieve mass production of CNTs. CNTs produced in high quantities are generally mixed or dissolved in functional powders, polymeric matrices, or molten metals. However, the fabrication of CNT-based high-quality nanocomposite remains a difficult issue because of the nonuniform dispersion of CNTs. Thus, the practical use of such composites is currently determined by the trade-off relationship between quality and cost. 

The direct growth of CNTs on different materials is another approach for fabricating composites. However, the growth of CNTs on powders is difficult to achieve in a conventional horizontal CVD system because of the confined contact between carbonaceous gas and the powder surface. To overcome this limitation, rotatable CVD (RCVD) has recently been used as an alternate methodology, yielding uniform growth of CNTs on the surface of functional particles [22,23,24,25]. Even though many studies have employed this method, a systematic and extensive study is required to understand the details of the RCVD system. 

In this study, we systematically investigated the effects of the process parameters of the RCVD system on the growth of CNTs on Al_2_O_3_ powder surfaces in achieving high uniformity at lower temperatures. The results show that the growth temperature could be reduced to 600 and optimized at 650 °C yielding uniform CNT growth with moderate quality.

## 2. Materials and Methods

### 2.1. Direct Synthesis of CNTs on Al_2_O_3_ Surface Using RCVD 

To synthesize CNTs on Al_2_O_3_ using RCVD, we first prepared catalytic iron nanoparticles from ferritin molecules because ferritin is composed of an iron compound core and a mineral shell. As previously reported [6,7,10], ferritin (Sigma Aldrich, Munich, Germany) was diluted with deionized water, and the concentration was adjusted to 0.5 mg/mL. Then, as-purchased Al_2_O_3_ powder (70 μm in diameter, 99% purity, Yee Young Cerachem, Seoul, Republic of Korea) was mixed in the ferritin solution at a ratio of 1 g:1 mL and magnetically stirred. The resulting solution was then poured into petri dish installed on a hot plate (80 °C) and dried to obtain a powder. We consecutively performed calcination of the ferritin-mixed Al_2_O_3_ powder at 550 °C for 10 min to remove the mineral shell of the ferritin. CNTs were grown on the Al_2_O_3_ powder surface using RCVD for 10 min and the furnace was air-cooled under Ar flow. The experimental procedure is schematically shown in Figure 1a.

The RCVD system was based on a conventional horizontal CVD chamber (5.08 cm in diameter, 150 cm long quartz tube) and a rotation function was added. The diameter of the central region was larger (7.62 cm) to confine the powder during operation as shown in Figure 1b. We measured the actual temperature in the quartz tube at 1 cm intervals using a K-type thermocouple and confirmed that the RCVD chamber had a uniform temperature region of 12 cm as indicated in Figure 1c. 

To grow CNTs, the prepared ferritin-mixed Al_2_O_3_ powder (1 g) was loaded in the central region of the quartz chamber. Subsequently, the chamber was purged twice below 5 × 10^−2^ Torr by using a mixture of Ar (900 sccm) and H_2_ (100 sccm) and heated to the desired growth temperature with 20 °C/min heating rate and a chamber rotation speed of 10 rpm. After stabilization for 10 min, CNT growth was initiated by changing the gas from Ar to ethylene (C_2_H_4_, 900 sccm). The growth temperature and time are critical factors affecting the amount of CNT grown and varied from 600 to 825 °C and from 10 to 30 min, respectively. Chamber rotation is also an important parameter to yield uniform growth of CNTs on the particle surface. Here, we adopted the rotation speed of 5 and 10 rpm. Experimental parameters to investigate the growth behavior of CNTs are summarized in Table 1. After the growth process, the chamber was furnace-cooled to ambient temperature under an Ar flow without rotation. To estimate the yield of CNTs, thermogravimetric analysis (TGA, SDT Q600, Perkin Elmer, Boston, MA, USA) was performed at a heating rate of 10 °C/min under air.

### 2.2. Characterization of Nanocomposites 

Field-emission scanning electron microscopy (FE-SEM, Hitachi S-4800, Tokyo, Japan) and transmission electron microscopy (TEM, Jeol JEM-2100F, Tokyo, Japan) were employed to characterize the structure of the obtained materials. Chemical element analyses were performed using energy-dispersive spectroscopy (EDS) attached to the SEM and TEM. TEM samples were prepared by dispersing the as-produced materials in ethanol via brief sonication and subsequent dropping of the solution onto a carbon-coated TEM grid. Raman spectroscopy (Horiba Aramis, Tokyo, Japan) was used for estimating the structural completeness of CNTs by comparing the peak intensities of a structural-disorder-induced peak (D-band, I_D_) at approximately 1350 cm^−1^ and the tangential stretching vibration mode of graphite (G-band, I_G_) at around 1590 cm^−1^ [26,27]. For the Raman analysis, the excitation wavelength of the laser was 532 nm and the spot size was 1 μm. An X-ray diffractometer (PANalytical X’Pert Pro, Eindhoven, The Netherlands) was used to investigate the evolution of the sample constituent. The analysis was performed using Cu-Kα radiation (λ = 1.5418 Å) with a step size of 0.1°. 

## 3. Results

### 3.1. Direct Synthesis of CNT on Al_2_O_3_ Powders Using a Horizontal CVD

Figure 2 shows the comparative experimental results for CNT growth on Al_2_O_3_ powder. The experiment was performed using a conventional CVD apparatus with a horizontal quartz tube (5.08 cm in diameter). The Al_2_O_3_ particles, shown in Figure 2a, are immersed in deionized water containing ferritin and poured on an SiO_2_-covered Si wafer. Figure 2b shows a low-magnification SEM image of CNTs uniformly grown on Al_2_O_3_ particles using C_2_H_4_ gas at 825 °C. Highly magnified SEM images show thin and long CNT structures as shown in Figure 2c,d. Figure 2e is the Raman profile of the as-purchased Al_2_O_3_ powder. The bands located at 415 and 644 cm^−1^ belong to the vibrational mode of *A_1g_* symmetry. Raman peaks located at around 382, 578, and 750 cm^−1^ were assigned to the *E_g_* vibration mode. XRD profile of the as-purchased Al_2_O_3_ powder, as shown in Figure 2f, reveals that it is α-Al_2_O_3_ with a rhombohedral crystal structure. 

Figure 2g shows a representative Raman spectrum obtained from the sample after the growth. The profile has vibrational bands typical of SWNTs, that is, radial breathing mode (RBM) below 200 cm^−1^, D-band, and G-band. In addition, peaks from Al_2_O_3_ were found at 382, 415, 575, 644, and 752 cm^−1^. The RBM peaks of single-walled CNTs (SWNTs) are related to their diameter by the following relation [26]: *ω* = 248/*d*(1)

Here, *ω* is the Raman frequency [cm^−1^] and *d* is the nanotube diameter [nm]. Thus, from the peak positions at 106, 122, 138, 160, and 168 cm^−1^, we estimated that the grown tubes were SWNTs with diameters of 2.33, 2.03, 1.79, 1.55, and 1.48 nm, respectively. Moreover, the splitting of the G-band into G^–^ and G^+^ peaks suggests that the grown tubes were SWNTs [28,29,30]. 

The XRD profile (Figure 2h) shows the diffraction peaks of the CNT and α-Al_2_O_3_ powder. From the diffraction angles from the CNTs, we can estimate the crystallite size (La) by using the Scherrer equation [31]: (2)$$La=\frac{K\lambda }{\beta cos\theta }$$
where La is the crystallite size, *K* is the Scherrer constant (0.94), λ is the wavelength of CuKα X-ray (0.154 nm) of the diffraction, β is the full width at half maxima (FWHM) of the XRD diffraction peak, 0.29 and 0.14 for conventional and RCVD systems, respectively. θ is the Bragg angle (21.5°) of the (100) plane of the tube structure. We finally obtained the values of 29.6 and 63.7 nm for conventional and RCVD systems, respectively. Here, we require more understanding of the values of crystal sizes because Bragg’s angles of the CNTs (26° for (002) and 43° for (100)) are superposed with those of α-Al_2_O_3_ particles. 

As mentioned earlier, the development of nanocomposites consisting of Al_2_O_3_ and CNTs has been extensively explored for use in heat-dissipation devices. Mechanical mixing has been predominantly reported in the literature [28]. As thermal transport through the interfaces between the ceramic particles and CNTs is important for the dissipation of heat, we attempted to grow CNTs directly on the ceramic powder as shown in Figure 2. However, the uniform reaction of the powder is limited in conventional horizontal and stationary CVD systems owing to the limited exposure to reactive gases during the CVD process. Therefore, we devised an RCVD apparatus and performed CNT growth on Al_2_O_3_ powder. 

### 3.2. One-Pot Synthesis of CNT on Alumina Powders Using Rotatable CVD 

#### 3.2.1. Effect of Growth Temperature

We investigated the effect of the growth temperature of CNTs on the final structures of the nanocomposites because low-temperature growth is vital for industrial applications. Figure 3 shows representative low- and high-magnification SEM images at different growth temperatures using the RCVD system. For this, the rotation speed was maintained at 10 rpm. As shown in Figure 3a, the Al_2_O_3_ particles had a roughened surface at a growth temperature of 600 °C. Such a change in the surface morphology of Al_2_O_3_ particles is believed to have been caused by their reduction during the heating, reduction, and growth stages. The magnified SEM image (Figure 3a’) captured from the roughened region showed many rectangular- or cube-shaped particles. We observed very short tubular structures approximately 33 nm in diameter (averaged from 15 tubes) and 330 nm in length (averaged from 5 tubes). When the growth temperature was 650 °C, the surface morphology was more uniform with smaller particles as shown in Figure 3b. In addition, cube-shaped particles were not observed. Instead, as shown in Figure 3a’, thick and thin tubular materials were observed on the Al_2_O_3_ surface. Similar growth behavior of tubes with varying diameters was also observed in the samples grown at 700 and 750 °C, as shown in Figure 3c,d, respectively. Here, most of the tubes were crumpled, and their length was still shorter than 1 μm. A drastic change in the structure and length of the tubes were observed in the samples grown at 800 and 825 °C, as shown in Figure 3e,f, respectively. In both cases, very thin tubes of several 10s of micrometers in length were observed to grow. Unlike the crumpled and thick tube morphologies that formed until 750 °C, the tubes grown at 800 °C and above were similar to few-walled CNTs (FWCNTs) with high structural quality. From the growth results using a conventional CVD chamber in Figure 2d–f, we believe that the produced tubes consisted of single-, double-, or triple-walled CNTs. In this study, the number of CNTs grown at 800 °C was larger than that at 825 °C sample. It must be mentioned that, at higher temperatures, the surfaces of the alumina particles were very smooth compared to those treated at lower temperatures. This leads us to hypothesize that the growth behavior of CNTs on Al_2_O_3_ particles at temperatures below 800 °C would be different from those above 800 °C; however, further investigation is required to confirm this. 

To investigate the change in the quality of the produced CNTs (including the crystal structures of Al_2_O_3_ and CNTs) with growth temperature, we performed Raman and XRD analyses, and the results are shown in Figure 4a,b, respectively. In the Raman spectra, some representative peaks of Al_2_O_3_ can be found at 382, 415, 575, 644, and 752 cm^−1^ from all samples. In addition, the D- and G-band peaks of the CNTs were also detected. The intensity ratio I_D_/I_G_, which gives information about the structural integrity of CNTs, increased from 0.70 to 1.92 with an increase in the growth temperature from 600 to 750 °C. Referring to the SEM images shown in Figure 3, an increase in the value of I_D_/I_G_ at a lower temperature (i.e., below 750 °C) may not imply an improvement in the structural integrity, especially in the case of multi-walled CNTs (MWCNTs). This is because we must consider the possibility of structural degradation caused by the collision or shearing of CNT-covered Al_2_O_3_ particles in the rotating chamber during the growth process, as previously observed in ball-milled CNTs [28,29,30]. Moreover, the formation of structural damage in CNTs, such as tube cutting and flattening, would be accelerated at elevated growth temperatures. Thus, the growth temperature with suppressed structural damage would be 650 °C. At higher temperatures, that is, at 800 and 825 °C, both samples show abruptly decreased I_D_/I_G_ values of 1.11 and 0.79, respectively. Even though the I_D_/I_G_ value of the sample grown at 825 °C was slightly low, based on the tube morphology (Figure 3e,f), we believed that the sample grown at 800 °C was also highly crystallized CNTs. In addition, because no RBM peaks were detected in the samples, we expect that the produced CNTs might be FWCNTs.

Figure 4b shows the XRD spectra containing the diffraction peaks of both α-Al_2_O_3_ powder and CNTs. As the diffraction peaks from all constituents did not broaden or shift, it can be concluded that the structural degradation was not pronounced, as suggested by Raman analyses. 

A comparison of the color of the products after the rotary CVD growth process is shown by a digital photograph in Figure 4c. As-purchased Al_2_O_3_ powder was white in color, but the sample grown at 600 °C was dark brown. The color changed to black in the sample grown at 650 °C and again to dark brown at 700 and 750 °C. We believe that the color might be related to the amount of CNTs in the composite. In addition, we could clearly confirm the uniformity of the products, which cannot be obtained using a conventional stationary CVD apparatus. Interestingly, the samples grown at 800 and 825 °C were light and dark brown, respectively, despite having long tubes with enhanced structural integrity. This can be attributed to the small diameters of the CNTs. Figure 4d shows the TGA analysis result of CNT/Al_2_O_3_ nanocomposites grown at various temperatures. From the values of weight loss shown as an inset in Figure 4d, we found that the weight of CNTs synthesized ranges from 0.1 to 1.3 mg from the 1 g of alumina powder. It is interesting that the variation of weight loss depending on growth temperatures is the same as the change of sample colors from brown to black in Figure 4c. Furthermore, the growth yield of CNTs was high at 700 and 750 °C and then abruptly decreased at 800 and 825 °C. Together with SEM (Figure 3) and Raman data (Figure 4a), we expect that the change in weight loss is closely related to the structure (diameter and length) and quality of the CNTs synthesized.

#### 3.2.2. Effect of Growth Time 

As the growth time increases, for instance from 10 to 20 and 30 min, the change in growth behavior is shown in Figure 5. The growth temperature was 650 °C with the chamber rotation of 10 rpm. In the case of a 10 min sample, thick and thin CNTs coexisted on the Al_2_O_3_ surface, shown in Figure 5a’ as mentioned earlier in Figure 3b’. Then, it is found that the growth area shown as bright gray is enlarged on the Al_2_O_3_ particles at the 20 min growth sample (Figure 5b). From the detailed view of Figure 5b’, as-grown CNTs have a relatively uniform diameter compared to the 10 min sample. Moreover, in the case of the 30 min sample, we can observe that CNTs are uniformly covered over the Al_2_O_3_ surface (Figure 5c) and CNTs are grown with high density as shown in Figure 5c’.

To investigate the structural integrity of the samples, Raman analysis was performed, and the results are shown in Figure 5d. The Raman profiles were typical of MWNTs [27], wherein the values of I_D_/I_G_ for growth time of 10, 20, and 30 min were 0.82, 1.57, and 2.20, respectively. From this, we surmise that longer growth times enhanced structural degradation, which could be due to the collision, abrasion, or attrition of CNTs/Al_2_O_3_ particles during the growth stage. Figure 5e shows XRD profiles depending on the growth time from 10 to 30 min at 650 °C. Different from the Raman analysis results, it was hard to observe the apparent difference in the diffraction patterns. This means that, from a point of structural quality, the optimization of the growth time should be based on the Raman results rather than XRD spectra. In addition, it is found that there is a tradeoff relationship between the growth amount and structural quality of the CNTs in RCVD growth system.

#### 3.2.3. Effect of Rotation Speed of CVD Chamber 

As mentioned earlier, process development yielding scalable and uniform growth is important for cost reduction and preservation of the functionality of the final products, especially in powder-based materials. In the rotary CVD process, the uniformity and scalability of the products are expected to be a function of the rotation speed. To verify this, we varied the rotation speed at 5 and 10 rpm and compared the growth behaviors, as shown in Figure 6. The growth was performed at 650 °C for 10 min.

Figure 6a shows the SEM image of Al_2_O_3_ particles treated with no chamber rotation, wherein it is difficult to observe any CNT. Even in the magnified image of Figure 6a’, very short tubes can be observed with a very low density. The results, therefore, clearly show the non-uniform growth of CNTs under stationary CVD conditions. When the chamber was rotated at 5 rpm (Figure 6b), the growth behavior was similar to that of the case without rotation. This implies that a rotation of 5 rpm did not have a significant effect. However, it was observed from the magnified view (Figure 6b’) that more tubular structures had grown on the Al_2_O_3_ surface. Figure 6c shows the effect of rotating the chamber at 10 rpm on CNT growth, wherein a relatively uniform growth of CNTs can be observed. Although we performed CNT growth at 20 rpm, the growth features are similar to those shown in Figure 3c’. Therefore, the optimized rotation chamber speed in this study was considered to be 10 rpm. However, to maximize the effect of chamber rotation, the rotation speed should be optimized because the growth results are largely affected by process parameters such as the feeding amount of powders, gas flow rate, and heating rate of the chamber.

Figure 6d shows the Raman spectra of the samples obtained from the three cases, wherein a minute increase in the I_D_/I_G_ value can be observed with increasing rotation speed, that is, it increased from 0.86 at 0 rpm to 0.87 and 0.97 at 5 and 10 rpm, respectively. As mentioned, some structural degradation is possible due to the collision or abrasion of CNTs/Al_2_O_3_ particles during chamber rotation. Figure 6e shows XRD profiles depending on the rotation speed from 0 to 10 rpm at 650 °C. It was also hard to observe the evident difference in the diffraction patterns. Further investigation is required to understand the structural degradation behavior during RCVD. 

## 4. Discussion on Growth Behavior of CNTs 

The above results show that the growth behavior is significantly affected by the growth temperature, growth time, and rotation speed of the CVD chamber. In general, the growth of CNTs can be explained by either a tip-growth or a base-growth model [6,7,8,9,10,11,12,13,14,15,16,32,33]. They are based on the binding forces between the CNTs, catalytic metals, and supporting substrates. The tip-growth model is used when the binding force between the CNTs and catalyst is higher than between the catalyst and substrate, otherwise, the base-growth model is used. In addition, the tip-growth model is frequently used in plasma-enhanced CVD growth of CNTs. In contrast, the base-growth model is more prevalent in thermal CVD cases. Most CNT growth recipes have adopted the use of a supporting layer, such as an e-beam-deposited or plasma-sputtered Al_2_O_3_ thin film [33,34,35]. Because the supporting layer pins catalytic nanoparticles (such as Fe_2_O_3_) by modifying the diffusivity of catalytic elements (Fe) on the film, the supporting layer having optimal thickness has generally been employed to control the density and diameter of CNTs [35,36,37,38,39]. 

Based on the results presented above and the ultimate goal of this study, 650 °C with 10 rpm can be said to be the optimized condition because it yields uniform growth of CNT with higher quality and moderate quantity on Al_2_O_3_ particles. Figure 7 shows the SEM image and corresponding two-dimensional EDS maps of the optimized sample. In the SEM image (Figure 7a), the catalytic particles are highlighted with black circles, and the same regions are encircled in white in the EDS map shown in Figure 7b, including the existence of Fe. It is, however, difficult to clearly discern the spatial distributions of aluminum, oxygen, and carbon, as displayed in Figure 7c–e, respectively. That said, the EDS mapping result implies that the white particles in the SEM image are Fe-based catalytic particles (such as Fe_2_O_3_ or Fe_3_O_4_) and are not part of the Al_2_O_3_ particles. Thus, the Fe particles can be attributed to ferritin.

Further investigation to understand the CNT growth behavior was performed using TEM. Figure 8a shows a representative image of relatively clean CNTs without apparent impurities. The magnified image in Figure 8b suggests that the CNTs have a relatively uniform diameter of approximately 15 nm (as highlighted by the black arrows). Figure 8c shows a high-magnification TEM image of the CNTs and nanoparticles. The structural quality was not high due to the crooked tube-wall structures, as also observed in the SEM images of Figure 5a’, Figure 6c’ and Figure 7a. Elemental mapping of the nanoparticles denoted by black circles was done using TEM, and the results are shown in Figure 8d,e. Herein, it can be observed that the particles consisted of iron and oxygen. Although the exact stoichiometry of the nanoparticles could not be determined, it is clear that they are catalytic nanoparticles of iron oxide.

## 5. Conclusions

One-pot fabrication of nanocomposites consisting of CNTs/Al_2_O_3_ particles was demonstrated using an RCVD system. CNTs were grown on the surface of Al_2_O_3_ after the attachment of ferritin-derived iron nanoparticles as a catalyst. We found that the CNTs could be grown at around 600 °C, and their amount increased with increasing growth time. In contrast to the limited growth of CNTs on the Al_2_O_3_ surface in the absence of chamber rotation, the presence of such a rotation allows uniform growth of CNTs over the particle surface. Within the scope of our investigation, we conclude the optimized conditions of CNT growth are 10 rpm at 650 °C for 10 min from a point of industrial application as filler materials in heat dissipation devices. That said, the process variables should be optimized depending on the amount, morphology, and size of the loading materials. Despite that, the present results demonstrate the versatility of the RCVD system for producing powder-based multidimensional composite materials with high uniformity.

## Figures and Tables

**Figure 1 materials-16-02735-f001:**
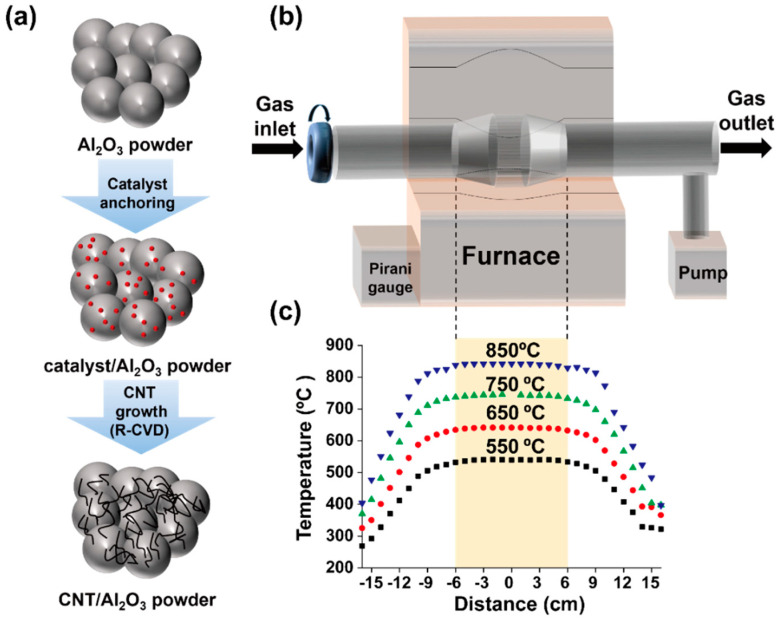
Schematic illustration of (**a**) experimental procedure and (**b**) rotatable CVD (RCVD) apparatus. (**c**) Temperature profiles in the RCVD chamber were measured with a thermocouple at various temperatures.

**Figure 2 materials-16-02735-f002:**
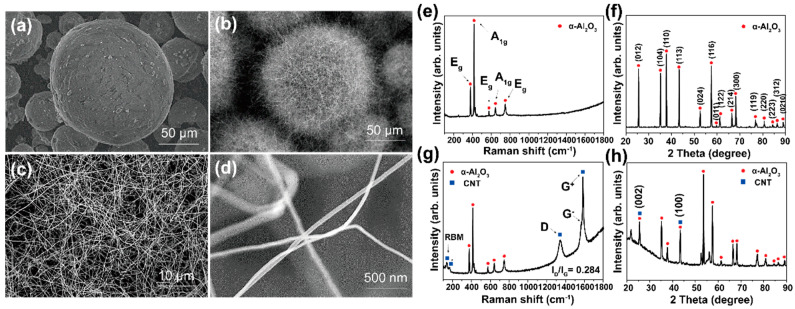
Low-magnification SEM images obtained (**a**) before and (**b**) after the CNT growth on Al_2_O_3_ powders, (**c**,**d**) show highly magnified SEM images. (**e**) Raman spectrum and (**f**) XRD profile from Al_2_O_3_ powders. (**g**) Raman spectrum and (**h**) XRD profile from CNT grown on Al_2_O_3_ powders at 825 °C using a conventional horizontal CVD chamber.

**Figure 3 materials-16-02735-f003:**
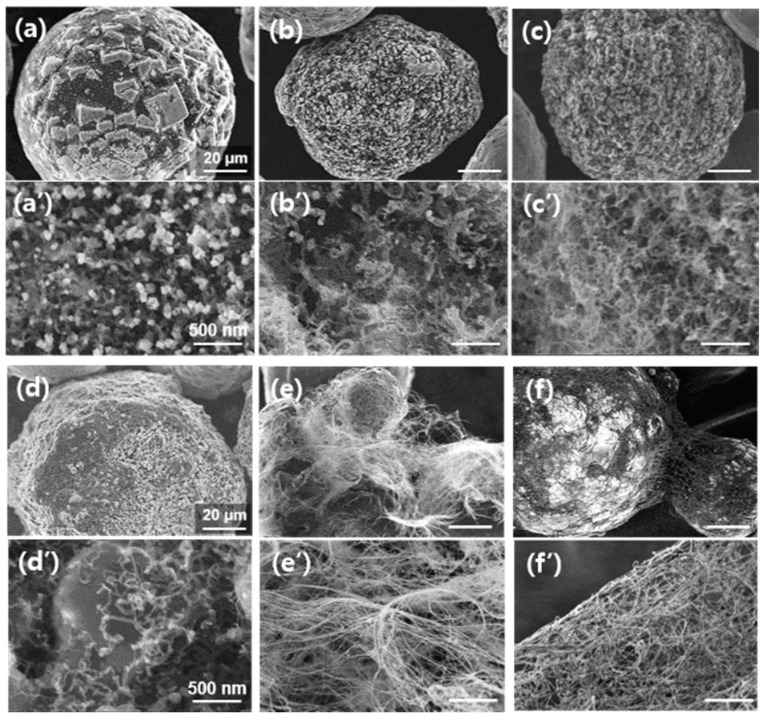
Low and high-magnification of SEM images showing CNTs directly grown on Al_2_O_3_ particles using RCVD system. Growth temperatures are (**a**,**a’**) 600, (**b**,**b’**) 650, (**c**,**c’**) 700, (**d**,**d’**) 750, (**e**,**e’**) 800, and (**f**,**f’**) 825 °C, respectively. Growth time and rotation speed are fixed at 10 min and 10 rpm, respectively.

**Figure 4 materials-16-02735-f004:**
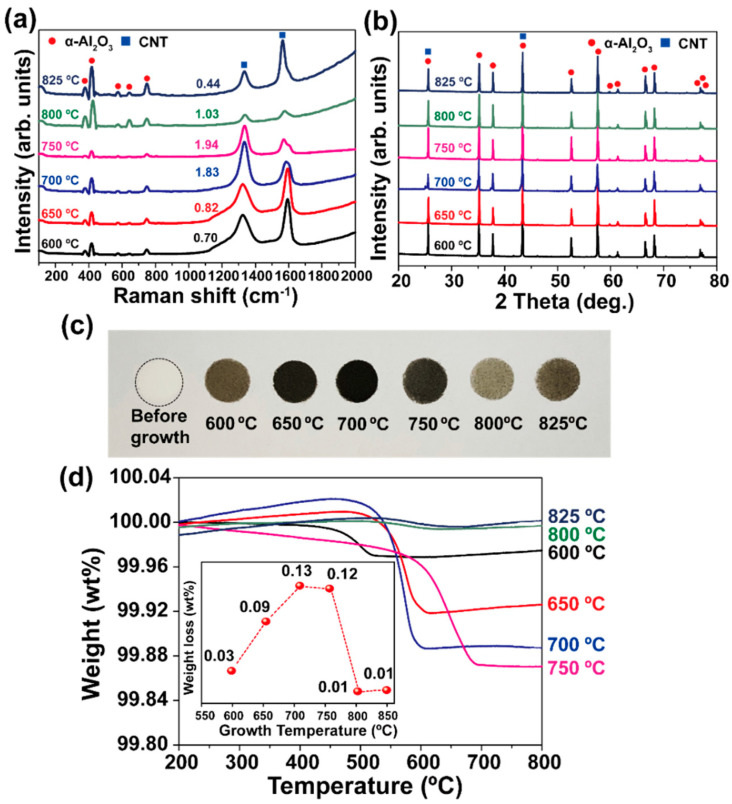
(**a**) Raman and (**b**) XRD profiles obtained from CNT/Al_2_O_3_ nanocomposites synthesized at various temperatures. (**c**) Digital photo showing color change of produced materials. The black line in before growth is to guide the eye. (**d**) TGA analysis result of CNT/Al_2_O_3_ nanocomposites grown at various temperatures. Inset shows a variation of weight loss depending on growth temperatures.

**Figure 5 materials-16-02735-f005:**
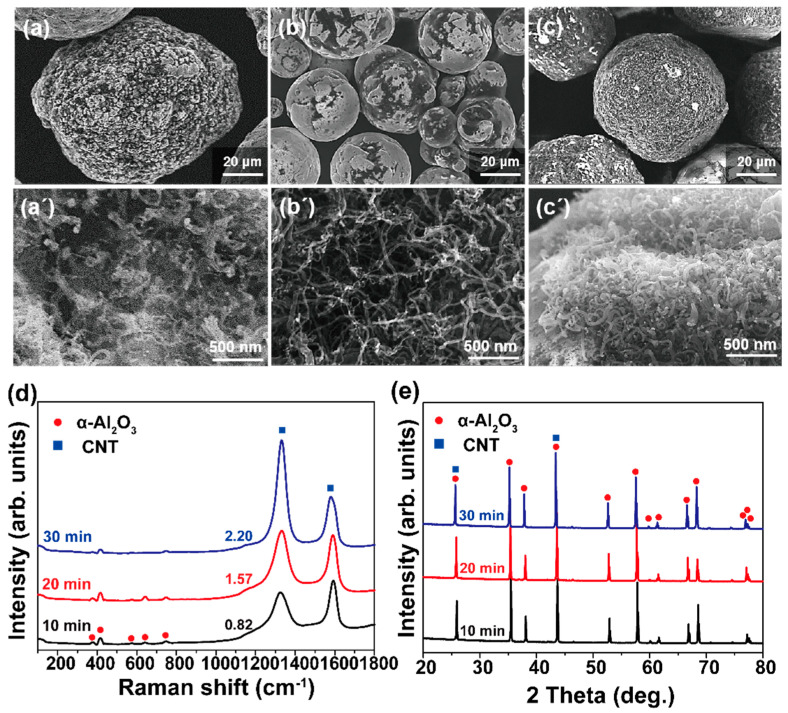
Low- and high-magnification SEM images showing CNTs directly grown on Al_2_O_3_ particles at 650 °C with 10 rpm of RCVD chamber. Growth time is (**a**,**a’**) 10, (**b**,**b’**) 20, (**c**,**c’**) 30 min. (**d**) Raman spectra and (**e**) XRD profiles obtained from (**a**–**c**) samples.

**Figure 6 materials-16-02735-f006:**
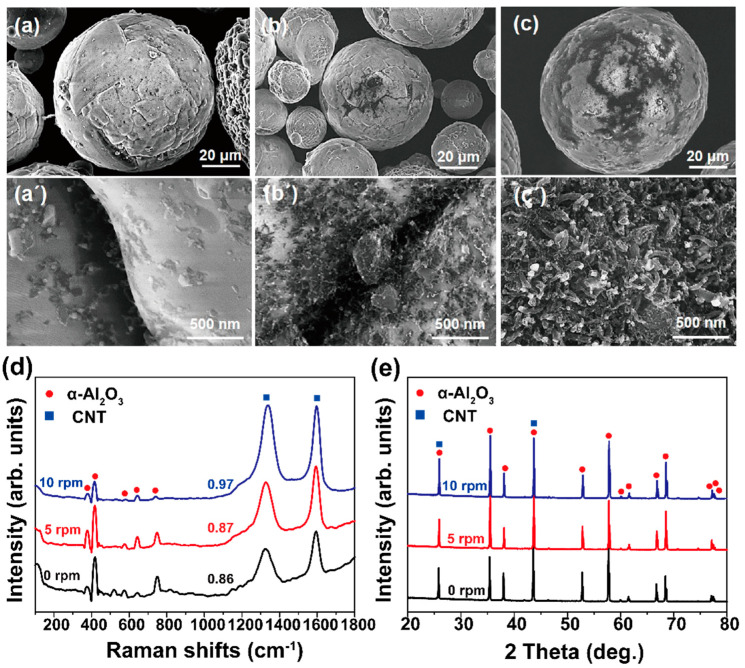
Low- and high-magnification SEM images showing CNTs directly grown on Al_2_O_3_ particles at 650 °C for 10 min. The rotation condition is (**a**,**a’**) 0, (**b**,**b’**) 5, (**c**,**c’**) 10 rpm. (**d**) Raman spectra and (**e**) XRD profiles obtained from (**a**–**c**) samples.

**Figure 7 materials-16-02735-f007:**
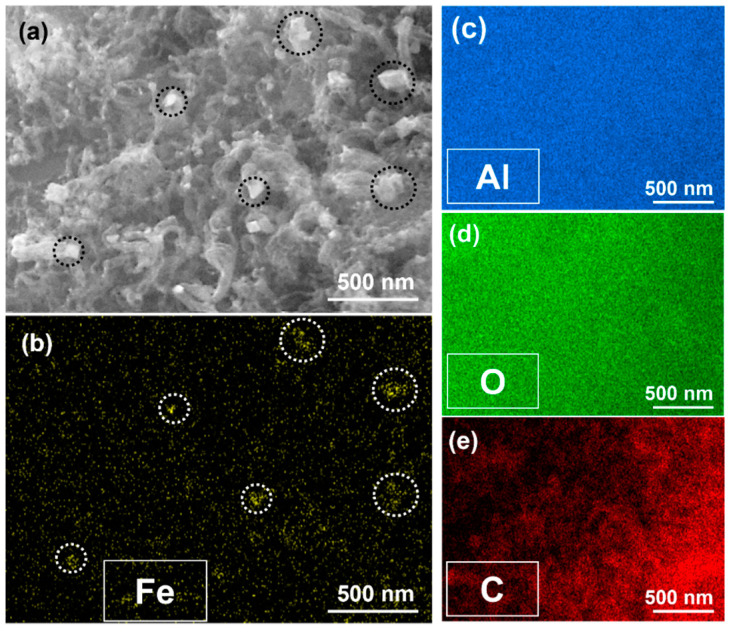
(**a**) SEM and corresponding EDS mapping images for (**b**) iron, (**c**) aluminum, (**d**) oxygen, and (**e**) carbon, respectively. The CNT was grown on Al_2_O_3_ at 650 °C for 10 min with 10 rpm of RCVD chamber. Black circles in (**a**) correspond to white circles in (**b**) indicating Fe particles.

**Figure 8 materials-16-02735-f008:**
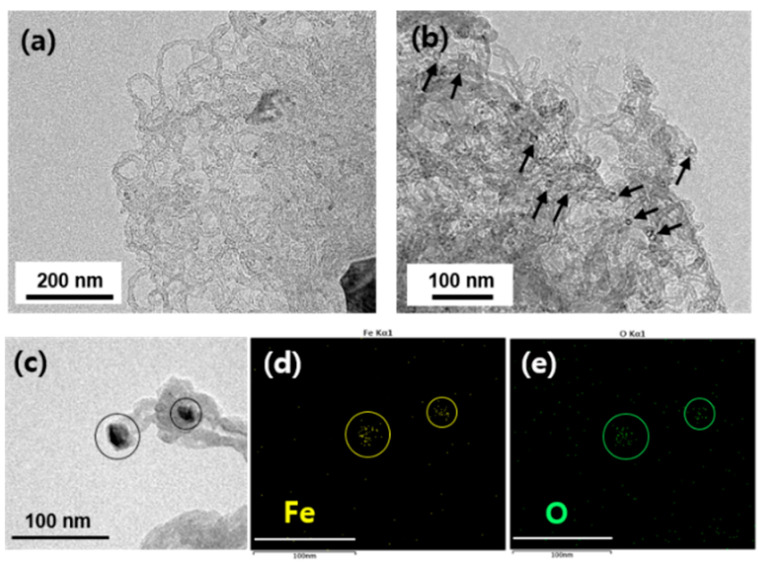
TEM images showing CNTs grown on Al_2_O_3_ at 650 °C for 10 min with 10 rpm of RCVD chamber. (**a**) A representative and (**b**) magnified images. Arrows in (**b**) indicate cross-sectional views of individual CNT. (**c**) A bright field TEM image of CNT. Corresponding EDS mapping results for (**d**) iron and (**e**) oxygen, respectively. Scale bars in (**d**,**e**) are 100 nm.

**Table 1 materials-16-02735-t001:** Experimental variables investigated in this study.

Experiment	Growth Temperature (°C)	Rotation Speed (rpm)	Growth Time (min)
**Growth temperature**	600, 650, 700, 750, 800, 825	10	10
**Rotation speed**	650	0, 5, 10	10
**Growth time**	650	10	10, 20, 30

## Data Availability

Not applicable.

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
