# Peer review of "One-Pot Fabrication of Nanocomposites Composed of Carbon Nanotubes and Alumina Powder Using a Rotatable Chemical Vapor Deposition System"

_materials, 2023, doi:10.3390/ma16072735_

Round 1

Reviewer 1 Report

Using rotatable chemical vapor deposition, Lee et al. report the growth of carbon nanotubes from Al2O3 supported Fe catalyst, and systematically investigate the effects of the main parameters on CNT growth. Interestingly, a faster rotation of the chamber allows for the uniform growth of CNT by the tip-growth mechanism. The manuscript could be publishable, but many issues should be addressed in the revised manuscript.

1.     Clearly, Fe nanoparticles are required to catalyze the growth of CNTs. The claim “Direct synthesis of CNTs on Al2O3 surface” is misleading.

2.     At a reaction temperature of 825 oC, single-walled carbon nanotubes were synthesized, and the authors use the RBM frequencies to estimate tube diameters. As Raman scattering is a resonant process, more laser wavelengths are suggested to use to characterize the nanotubes.

3.     Meanwhile, TEM is encourage to analyze the structure and morphology of the products.

4.     The authors state in the conclusion that “the optimized conditions for CNT growth are 10 rpm at 650 °C for 10 min”. What are the criteria? The yield, quality or something else?

5.     To estimate the yield of CNTs, thermal gravimetric analysis should be presented.

Author Response

We appreciate for your kind notification of comments on our manuscript (materials-2261903) ‘One-pot fabrication of nanocomposites composed of carbon nanotubes and alumina powder using a rotatable chemical vapor deposition system’.

It is our great pleasure that the reviewer suggested very constructive comments and we revised the manuscript according to the comments.

Finally, we would be glad if the revised manuscript will be suitable for publication and also thank for your kind cooperation in advance.

Reviewer 2 Report

This work reports the study of one-pot fabrication of nanocomposites α-Al2O3/CNT using a rotatable CVD system. After review, I would like to express the following comments; before acceptation, some comments need to be addressed:

1.-Line 82, instead of 500 0C, it should be 600 0C.

2.- Add a brief explanation of the experimental design of the effects of temperature, rotation speed and growth time on CNT synthesis in the RCVD system, in section 2 (materials and methods).

3.- It’s suggested to add a table where the experimental design of RCVD system is explained in section 2 (materials and methods). An example could be as follows:

Experiment

Growth temperature

(0C)

Ration speed

(rpm)

Growth time

(min)

Growth temperature effect

600, 650, 700, 750, 800, 850

10 *

10 *

Rotation speed effect

650 *

5, 10

10 *

Growth time effect

650 *

10 *

10, 20, 30

* Fixed parameter

4.- Regarding the direct synthesis of the α-Al2O3/CNT compound in a traditional CVD system, we assume that it was done for comparison purposes, something that is not very clear in the work. The test temperature was 8250C, but it doesn't say the growth time (rotation speed is zero). A brief explanation of the synthesis should also be given in the materials and methods section.

5.- Indicate the vibrational modes in Raman Spectra of α-Al2O3  in Figure 2b.

6.- Indicate the vibrational modes in Raman Spectra of CNT in  Figure 2f.

7.-Indicate CNTs crystal planes in Figure 2g.

8. It could be compare the crystal size of the CNT obtained from the diffractogram (using Scherrer or Rietveld refinement), with the diameters obtained from the RBM modes of the CNT in the sample by direct CVD.

9. Calculate the ID/IG ratio of the CNTs synthesized by traditional CVD, for comparative purposes with the nanotubes synthesized by RCVD.

10. line 165 and 194, I think it should be FWCNTs, instead of FWNTs.

11. Indicate in Figure 4b which diffractogram corresponds to each synthesis temperature.

12. Diffractograms of the synthesized samples should be included to observe the effect of growth time and rotation.

13. Since RBM modes are not visible in the samples synthesized by RCVD, they could be compared with the CNTs synthesized with traditional CVD using the crystal sizes obtained from the XRD diffractograms.

14. We assume that the conditions used in the traditional CVD (Growth temperature: 8250C, growth time: ?) are optimal, where CNTs of the best quality and size (diameter and length) were obtained. As for the amount obtained per test, does this improvement with the RCVD system? It would still be interesting to know this detail, and compare diameter, length and quality of CNTs.

Author Response

(The authors gave the same response as above.)

Reviewer 3 Report

Authors present the systematic investigation of the effects on the process parameters for the rotatable chemical vapor deposition (RCVD) system applied for the growth of carbon nanotubes (CNTs) on Al2O3 powder surface.  The rotation speed and temperature are successfully optimised for the CNT preparation via RCVD method. Thorough physical characterization is provided for the CNTs prepared in different conditions to prove the success of the optimization. This is an original and novel work that should appeal to the reader of Materials journal. Prior to publication some of the issues pointed out by the reviewer should be addressed in the manuscript.

Reviewer concerns:

1   1. Authors should state more clearly the novelty and benefits of their RCVD system compared to the conventional CVD system for CNT preparation. Currently, the reviewer could only find a certain benefit of Si-wafer being not needed to support the Al2O3 powder directly in the gas flow. This was rather hard to find and is definitely not the only benefit for the RCVD system.

2       2. Page 3, Figure 2, (b,c,f,g) numbers on x-axis are too small for reading, also the XRD peak designations in (c) are rather small.

3      3. Page 4, line 115, “The magnified SEM image in Fig. 2(e) reveals the uniform growth of high-quality CNTs.” Could the authors please provide also the high-resolution SEM image for the CNTs shown in Figs 2 (d, e)? For example, similar to the ones in Fig. 3 with 500 nm scale bar.

4      4. Page 4, line 126, “Thus, from the peak positions at 106, 122, 138, 160, and 168 cm-1, we estimated that the grown tubes were SWNTs with diameters of 2.33, 2.03, 1.79, 1.55, and 1.48 nm, respectively” Could the authors provide any evidence (e.g. TEM images) to clarify if these calculated diameters could be verified?

5      5. Page 5, line 154, “We observed very short tubular structures with approximately 33 nm in diameters (averaged from 15 tubes) and 330 nm in length (averaged from 5 tubes).” The reviewer could not understand clearly which SEM images was used to conclude this observation?

6      6. Page 6, line 223, “The Raman profiles were typical of MWNTs” A reference should be provided for this claim.

7      7. Page 9, Figure 7, a scale bar is missing from the figure.

Author Response

(The authors gave the same response as above.)

Round 2

Reviewer 1 Report

To address the previous comments, the authors have performed more characterizations on the carbon nanotube products, which I really appreciate. However, based on the new data, I find that the carbon nanotube yield and quality are rather poor. Besides, from the TGA profile, carbon nanotubes grown at 750 oC or 700 oC degree are the best in terms of yield and quality. However, the authors still claim that CNTs grown at 650 oC are the optimized conditions. Therefore, I could not recommend the manuscript for publication.

Author Response

Response 1: We appreciate that reviewer mentioned very important point. We aware that the present version should be revised to explain our experimental purpose and results clearly in order to prevent confusion of readers. First, we totally agree with the comment that the best growth temperature was 700 oC in terms of production yield based on the TGA data [Fig. 4(d)]. Thus, we revised the manuscript to express that point.

However, according to the values of ID/IG in Raman analysis [Fig. 4(a)], we thought the the quality of CNTs grown at 650 oC was better than that of CNTs grown at 700 and 750 oC. Concerning the production yield of CNTs, the weight loss in TGA data was very low. But, this result was caused by the fact that the alumina powders used as growth template are much heavier than CNTs.

In addition, the ultimate goal of the study is to fabricate CNT/alumina composite to be used in heat dissipation application. Furthermore, the uniform growth of short CNTs on alumina powder (filler) is preferable to have superior property as thermal interface materials.

Based on the above background, we described the optimized growth temperature was 650 oC showing uniform growth of CNTs with higher quality and shorter length within the range of this study.

Finally, we modified the manuscript to reflect the comment of the reviewer.

  • Page 7, line 246
  • Page 10, line 325
  • Page 11, line 362

Round 3

Reviewer 1 Report

The manuscript has been sufficiently improved to warrant publication in Materials